# Review of Fibromyalgia (FM) Syndrome Treatments

**DOI:** 10.3390/ijerph191912106

**Published:** 2022-09-24

**Authors:** Liraz Cohen-Biton, Dan Buskila, Rachel Nissanholtz-Gannot

**Affiliations:** 1Ariel University in Samaria, Ariel 4076414, Israel; 2Ben-Gurion University of the Negev, Beer Sheva 8443944, Israel

**Keywords:** fibromyalgia, cognitive behavioral therapies, cannabinoids, physical activity

## Abstract

Background: Fibromyalgia (FM) is a disease characterized by widespread musculoskeletal chronic pain that impairs the patient’s quality of life and is considered a somatization disorder. The symptoms of the disease also affect the patient mentally, mainly since invisible pain is the only thing that indicates its existence. A typical symptom that characterizes FM patients is the lack of acceptance of the disease since its pathophysiology is not elucidated, hence the deficiencies in its management, or rather, cognitively, the belief that there is no disease to manage. The current paper aims to shed light on the new treatment methods at a holistic level, that is, cognitive, physical, and pharmacological therapies. Method: A literature review was carried out that discusses treatment methods that help alleviate the pain, accept it, and manage the symptoms of the disease. Results: FM symptoms can be treated by taking a broad view of treatment that will include a response to the mind through pain management, response to the body through physical activity, and response to the pain through pharmacological treatment. Conclusions: Today, there is an evolutionary view that accepts FM and chronic pain diseases as syndromes in which the pain is the disease; therefore, the response to this disease can be applied through three channels: physical, bodily, and mental.

## 1. Overview on Fibromyalgia

Fibromyalgia (FM) is on the spectrum of syndromes that lack a precise classification, and it is present in 2–4% of the population [1,2]. FM is characterized by widespread musculoskeletal chronic pain and associated fatigue, sleep disturbances, and other cognitive and somatic symptoms [3,4]. It is often considered part of the general view of somatic and medically unexplained functional somatic syndromes or as a somatization disorder [1]. There is often an overlap between diagnosis and classification for case definitions of FM with various somatization disorders since patients with FM share symptoms with other functional somatic problems, including issues of myalgias, arthralgias, fatigue, and sleep disorders [5]. There are similarities to neuropathic pain in clinical findings, pathophysiology, and neuropharmacology [1,2]. Although FM is not a musculoskeletal disease, most symptoms manifest at this level [6,7,8].

Chronic pain is defined as pain that lasts or recurs for more than three months. In chronic pain syndromes, the pain is often the sole or a leading complaint and needs to be explicitly addressed. In syndromes such as FM or situations of nonspecific low-back pain, chronic pain may be perceived as a disease in itself [9] and described as “chronic primary pain”. Though its exact pathogenesis is still unclear, such pain that persists despite adequate treatment and in the absence of any sign of inflammation has led the research body to look for evidence of central sensitivity [10]. It is now clear that FM is involved with neural over-sensitization and decreased conditioned pain modulation (CPM) [11]. There is disagreement between different researchers regarding the presence of physical damage by FM. Trouvin et al. [12] argued that since FM is associated with chronic pain without any visible damage to peripheral tissue, the idea of nociplastic pain resulting from altered nociception was born. This means that any changes in the nervous system (both peripheral and central) can lead to pain even without any evidence of threat or tissue damage or evidence of any causative disease or lesion of the somatosensory system. In this hyperexcited state, when persistent nociceptive input leads to increased excitability in the dorsal horn neurons of the spinal cord, the spinal cord neurons produce an enhanced responsiveness to noxious stimulations and even to formerly innocuous stimulations [13]. On the other hand, in a systematic review, Galosi et al. [14] found that among FM patients, there are signs of damage to small fibers, which may lead to pain and autonomic symptoms. The estimated prevalence of small fiber somatic impairment was 49%. However, according to the authors, the heterogeneity and inconsistency between studies challenge the precise role of small fiber damage in FM symptoms. Other findings that showed neurological damage have emerged from various studies [15,16,17,18], which lead to the conclusion that FM is considered a neurological disorder. Many diseases have internal damage, but in most cases, they have visible physical signs, particularly when it comes to chronic diseases. In contrast, FM is a disease without visible symptoms. That is, for the immediate environment, family members, friends, employers, and even the patients themselves, there are no visible findings. Moreover, the diagnosis of internal nerve damage is only made after comprehensive tests, as the above studies have shown. In most cases, women who suffer from chronic fatigue and pain consult their family doctor. Additionally, even after they are diagnosed with FM, in the ordinary national health system, they do not undergo further comprehensive internal tests. That is, the institutionalized medical system does not prompt more tests other than the diagnosis of FM, and the interest in investigating further is an academic interest, as the above authors have demonstrated. To the population that surrounds FM patients, the disease is completely hidden, which leads to skepticism regarding its existence. The research by Cohen-Biton et al. [7,8] found that FM patients face both a lack of self-acceptance towards the disease and a lack of acceptance by others in their environment (family, friends, colleagues). Hence, this paper aims to provide new insights into treatment methods and to provide answers within the emotional realm that underlies coping with FM.

## 2. Material and Methods

### Search Strategy

We conducted a systematic review of the literature using NIH, PubMed, PMID, BMC, Medline, PsycINFO, and WebMD. In the first phase, we formulated the purpose of the study: to examine the new ways of treating fibromyalgia. We used the phrases: “fibromyalgia”, “chronic pain”, “chronic musculoskeletal pain”, “widespread musculoskeletal pain”, “therapeutic interventions for fibromyalgia”, and “fibromyalgia treatment”. The inclusion criteria were articles from research libraries in clinical medicine. The exclusion criteria were pre-2010 research papers and articles not from clinical medicine journals. After formulating the chapters’ headings, we reviewed about 100 articles in the various areas of the research. We included in the corpus only articles that described recent studies. After performing data extraction and filtering to detect duplicate results, we synthesized the information and analysis of the findings.

The next chapter will present an overview of the currently leading treatments.

## 3. Findings

### 3.1. Myofascial Physical Therapy

A term that has come up in the context of FM is “facilitation phenomenon”, which explains musculoskeletal pain events, particularly myofascial pain. This means that neural structures may become hyperreactive at the spinal or paraspinal tissue level (“segmental facilitation”). Points of hyperreactivity were defined as “tender points” in ligaments, tendons, or periosteal tissue and in muscle or fascia [19]. The literature identified 18 allogenic points when fibromyalgia involves pain caused by digital pressure on at least 11 out of 18 points [20]. In their systematic literature review on manual therapy in FM, Schulze et al. [21] found that myofascial release was the most used modality. Myofascial tissues are the tough membranes that wrap, connect, and support the body’s muscles [22]. Myofascial release is a treatment technique that is also used in massage when the therapist/masseur focuses on the pains that apparently stem from myofascial tissues [23]. In a study of 86 FM patients, 10 patients received myofascial release modalities and showed marked improvement in “tender points”, physical function, and clinical severity. Six months after the intervention, the patients who received myofascial release showed a significant reduction in “tender points”, physical function, and clinical severity [24].

### 3.2. Self-Management Skills

Successful pain management incorporates the patient’s education in self-management skills [25]. There is evidence showing that support for self-management does work. Supporting people to take care of themselves can improve their motivation, eating, and exercise routine as well as their symptoms and clinical outcomes [26]. A systematic review found a significant positive clinical effect of self-management programs for pain relief [27].

In an observational interview study conducted among FM patients who received instruction books and guidance for exercises, the authors sought to examine the effect of the clinically developed Fibromyalgia Self-Management Program (FSMP) [28]. This program is a nonpharmacological, multidisciplinary education group intervention seeking to provide patient-focused education and practice counseling to develop core self-management skills. The observations lasted for about six weeks. The findings showed that the FSMP sufficiently mapped the objectives and planning using five specific techniques learned from the instruction books. Qualitative interviews showed that these techniques are a powerful tool to facilitate change in participants’ behavior. The patients mainly employed the goal-setting technique, which is a strategy that has been successfully used in self-management interventions for long-term musculoskeletal conditions and pain management programs. At the end of the program, the participants expressed a desire to continue to conduct pain management activities such as aquatic physiotherapy or hydrotherapy [28].

### 3.3. Cognitive Behavioral Therapy (CBT)

The cognitive behavioral therapy (CBT) method was developed in the 1970s by an American psychiatrist named Dr. Aaron Beck. His main argument was that behavioral therapy can be effective in depression and coping with other problems, and that the desired change can be created by acquiring self-control skills and social skills and changing limiting thoughts and conditioning that create and intensify the problem. CBT is an approach that focuses on identifying patients’ difficulties and problems and responses to those difficulties and problems, i.e., the way the patient tends to respond. Once the problem and the typical form of response have been identified, a conceptualization is created, and the goals and treatment plan are determined, with the goal being to create alternative mental and behavioral patterns based on facts rather than interpretation: in other words, first a change of mind, and only then a change of behavior. To create change and overcome difficulties or mental problems, there is no need to be grounded in the past but to focus on the present. Therefore, therapy allows patients to change their thought patterns in the “here and now”, affecting their reactions [29,30]. Various systematic reviews have [31,32] suggested that CBT is an effective approach to chronic pain management. 

A standard method of CBT treatment is mindfulness. Since FM patients usually suffer from sleep problems, Park et al. [33] sought to examine the relationship between mindfulness and sleep in FM patients and how this relationship may be mediated by depression, anxiety, and pain disorders. The results showed that high mindfulness was associated with better sleep quality, a reduction in sleep disorders and pain, and a reduction in depression disorders. Cejudo et al. [34] aimed to evaluate the effects of a mindfulness-based intervention (MBI) on the improvement of subjective well-being, trait emotional intelligence (TEI), mental health, and resilience among women with FM. The authors conducted an intervention and follow-up evaluation program six months after the end of the intervention. The participants were divided into an experimental group and a control group. The findings showed that MBI showed statistically significant differences in satisfaction with life (SWL), positive affect (PA), mental health, and resilience.

In a study that was conducted aiming to analyze the cost effectiveness of treatment among FM patients in Spain [35], the authors developed an experimental intervention method of mindfulness-based stress reduction (MBSR) as an adjunct to treatment-as-usual (TAU). The results indicated that MBSR treatment achieved a significant cost reduction, especially indirect costs, and primary health care. In other words, the additional MBSR treatment was shown to be effective over TAU only.

Although the pathogenesis of FM is unknown, imaging has shown that alterations in central sensitization, involving an imbalance of brain-derived neurotrophic factor (BDNF) and inflammatory biomarkers, appear to be implicated [36,37]. Montero-Marin et al. [36] sought to assess the impact of attachment-based compassion therapy (ABCT) on BDNF levels and the levels of inflammatory markers, namely, TNF-α, IL-6, IL-10, and the C-reactive protein (CRP). BDNF is known to play a crucial role in various neuroplastic processes, including pain modulation, pain transduction, nociception, and hyperalgesia, which are all altered in FM [37]. Montero-Marin et al. [36] analyzed whether biomarkers are mediating/moderating in FM patients. The randomized controlled trial (RCT) sample consisted of FM patients who were divided into the ABCT group and the relaxation therapy (RT) group. The results showed significant improvements in the ABCT group Fibromyalgia Impact Questionnaire (FIQ) and a decrease in BDNF, CRP, and pro-inflammatory composite compared to the RT group. ABCT appears to reduce BDNF and have anti-inflammatory effects in FM patients.

Another RCT study in Spain by Sanabria-Mazo et al. [37] based on the RT method of Montero-Marin et al. [36] compared RT to Mindfulness plus Amygdala and Insula Retraining (MAIR) among FM patients. The study was based on the practice of mindfulness-based stress reduction (MBSR) in addition to the techniques of amygdala and insula retraining (AIR). The intervention in this method focused on improving skills and strategies for dealing with stressful situations. The study found that compared to RT, MAIR, in addition to TAU, is an effective intervention that has been proven to improve a wide range of effects: functional impairment, clinical severity, and quality of life. It is in conjunction with psychopathology cognitive processes, such as mindfulness and self-compassion. The beneficial effects of MAIR remained significant even after three months of follow-up and even improved in terms of clinical severity, perceived health, pain catastrophizing, and psychological flexibility. In addition, a significant decrease in BDNF levels in the MAIR group was observed after the intervention. It should be noted that despite the excellent clinical improvements and higher BDNF reductions in MAIR compared to RT, no significant changes were found with respect to immune-inflammatory variables. The results indicated an excellent potential effect of the combination of mindfulness and AIR. However, there is room for more comprehensive studies [37].

### 3.4. Physical Activity

A systematic review conducted between 2007 and 2011 found that exercise is generally recommended in treating people with FM, and interest in examining the benefits of exercise for these patients has increased significantly in recent decades. The research body supports aerobic and power training and other forms of exercise to improve fitness and physical function, reduce FM symptoms, and improve the quality of life [38]. However, exacerbating symptoms following exercise is a typical phenomenon for FM patients. These aggravations often cause them to develop a fear of performing a body movement or exercise and, as a result, to develop avoidance behavior toward physical activity. However, exercise is essential for successful treatment among FM patients [39,40,41]. Thus, customized CBT and exercise training are the most promising strategies for treating fear of movement and avoidance behavior toward physical activity in FM patients [39,40].

Systematic reviews and meta-analyses have found strong evidence that physical exercise reduces the pain intensity and significantly improves FM patients’ quality of life and physical and psychological functioning [42]. A systematic review designed to determine what types of exercise programs for FM patients were developed and their effects and benefits on pain and quality of life found that exercises focusing primarily on dance activities, water activities, body–mind work, fitness, and stretching were the most effective. The study showed that the level of pain was reduced between 10 and 44.2% after a multidisciplinary intervention program. In addition, the negative effect of the disease on quality of life also decreased between 5.3% and 17.9% while improving the symptoms of these patients [43]. Norouzi et al. [44] compared the effects of aerobic exercise and Zumba dancing on working memory, motor function, and depressive symptoms in women with FM and found a significant improvement in these parameters. Aquatic physiotherapy and a health education intervention program among FM patients showed a statistically significant reduction in all FM measures: fatigue, functional capability, anxiety, depression, and quality of sleep, except for a reduction in pain [45].

In another trial that examined the effects of exercise on symptoms and function in adults with FM, it was found that participants increased their average daily step count by 54% in the 6 min walk test, improved their overall function by 18%, and reduced their pain by 35% [46].

One of the issues that have come up in exercise among FM patients is persistent inactivity. Fontaine et al. [46] found that the significant improvement achieved during participation in an intervention program that included physical activity was not maintained over time. A six-month combined training program was conducted in a three-year longitudinal study among FM patients. The study found that a long-term training program can produce immediate health improvements in women with FM and that the benefits achieved in regular training can be maintained for 30 months [47].

### 3.5. Pharmacological Treatment

Randomized controlled trials (RCTs) have shown that a wide range of different medications have treated FM, including antidepressants, opioids, nonsteroidal anti-inflammatory drugs, sedatives, muscle relaxants, and antiepileptic drugs [48]. However, it is difficult to point the finger at fibromyalgia patients’ most common and prevailing type of drug treatment. The FDA has approved three drugs to treat FM: the antidepressants duloxetine (Cymbalta) and milnacipran (Savella), plus the anti-seizure medicine pregabalin (Lyrica) [49]. However, a large body of clinical trials has also tried other types of antidepressants to treat various chronic pain symptoms, including FM. In a study by Sumpton and Moulin [2], the authors found that tricyclic antidepressants, serotonin–norepinephrine reuptake inhibitors (SNRIs) (duloxetine and milnacipran), and α2-δ ligands (gabapentin and pregabalin) are effective in reducing FM-associated pain by ≥30%. Amitriptyline was also associated with improvement in pain fatigue, sleep, and quality of life [50,51,52].

### 3.6. Cannabinoids

There is mounting evidence supporting cannabis use in chronic pain conditions [53]. An in-depth review has shown that cannabis has proven benefits in treating chronic pain [54]. However, there is only a handful of randomized trials in FM patients, even though cannabis was associated with beneficial effects on some FM symptoms [55]. With the rising global trend of cannabis use in chronic pain emphasizing FM, evidence is being gathered that supports cannabis use among FM patients [53].

There are two major active components in cannabinoids, the chemicals that are derived from Cannabis: Δ^9^-tetrahydrocannabinol (Δ^9^-THC) and cannabidiol (CBD). THC is a psychoactive component that affects pain (as well as emotions) and works through CB_1_ and CB_2_ receptors [56]. The CB_1_ receptor binds the THC and mediates most of the CNS effects of THC [57]. CBD has anti-inflammatory and analgesic traits. The THC:CBD ratio, therefore, determines the product’s overall effect [56]. CB_1_ cannabinoid receptors are found predominantly in the central nervous system (CNS) and peripheral nervous system (PNS). Their agonists act along sensory pathways as pain modulators [58]. A study conducted among FM patients found an improvement within two hours in visual analog scale (VAS) scores, a reduction in pain and stiffness, an enhancement of relaxation, and an increased feeling of well-being [55]. In a longitudinal six-month study conducted on a relatively large cohort of FM patients (*n* = 367), there was a significant decrease in the average intensity of pain sleep disturbances, and symptoms associated with depression [59]. Canadian FM Management Guidelines also suggest cannabinoids as a therapeutic option for FM patients with major sleep disorders [60].

## 4. Conclusions

Although FM syndrome lacks a precise classification, it is a disease that significantly impairs the quality of life. As a somatic disease without visible evidence of threat or tissue damage or evidence of any causative disease of the somatosensory system, the body of research and the clinical world are increasingly accepting the recognition of a disease in which the nociplastic pain results from altered nociception. In other words, the pain is the disease itself. According to Raffaeli and Arnaudo [61], there is evidence that supports the conceptualization of pain as a disease, and therefore it is believed that a clear definition is needed, due to the enormous burden of this condition. Recognizing pain as a pathological condition can raise awareness of a global health problem as well as exploring new avenues of treatment. Thus, the treatment of the disease should address three channels: The first is the treatment of the pain itself, i.e., providing pain relief. The second channel should address the physical aspect by providing physical therapy to strengthen the body. The third channel addresses the disease’s mental element and psychosomatic structure. Our review discussed solutions employed today, and it is worth noting that other treatments have not yet crossed the threshold of human trials that are designed to address diseases characterized by chronic pain. These solutions encompass the physical realm with pharmacological and cannabinoid treatments and treatments designed to address the mental aspect of the disease, such as the development of self-management skills and CBT, while other treatments are designed to strengthen the body to reduce the pain and include physical activities and myofascial physical therapy. In the modern world with advanced technology, fibromyalgia is still a “transparent disease”, that is, a disease that has no visible symptoms, and in fact, the bystander needs to take the patients’ words that they are sick. Thus, designing a holistic body–mind–soul therapy can provide an appropriate response and promote women with fibromyalgia’s health.

## Data Availability

Not applicable.

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
