# Peer review of "Review of Fibromyalgia (FM) Syndrome Treatments"

_ijerph, 2022, doi:10.3390/ijerph191912106_

Round 1
Reviewer 1 Report
Could you consider changing your Title to include the word review.
Apart from your discussion on cannabis I didn’t see much new in the article.
Abstract ln 9
You state that FM is considered a somatisation disorder.
Surley this is now “old” thinking? In your article you provide evidence that it does appear to have an organic basis.
If you want to say that it is a somatization disorder perhaps try “some authors have classified FM this way” and give a reference.
Incidentally I understand that the new psychiatric term is “Somatoform disorder”.
In line 10 and 11 you introduce the term “transparency” is this the first time this word has been used to describe FM?
Also “invisible pain” as all pain is invisible ie it cant be accurately measured as it’s a self report or a behavioural response Im not sure it’s a good phrase to use here unless you justify later in the text.
Line 17 suggests that the only way to treat pain is with drugs. Is this true?
Line 20 Its not clear to me that there are three channels. Certainly physical and psychological (mental). What is bodily?
Overview
Ln28 You quote ref 1 to support your statement about somatization. This in my opinion is not correct Ref 1 puts somatic pain disorder in context of a range of considerations for the understanding of FM.
In line 41 you start giving evidence against somatization. This is confusing.
Line 51 you introduce the word “transparent” is this a good word to use here? Has it been used before to describe any diseases? Using the word women in the same sentence suggests it might only be transparent in women.
Materials
Ln 66 can you be more specific than “about” The details of your review are not clear. How did you select “Chapter Headings”? which post 2010 papers did you not review? Were you biased?
Lin7. “Next paragraph” might be a better word than “next chapter”
Findings ln83
You report on ref 19 stating 10 out of 86 FM patients responded to myofascial therapy, what does this mean? There are myofascial subsets of FM patients or it was just natural history of the fluctuating course of FM?
Ln 107 what is the difference between aquatic physiotherapy and hydrotherapy?
Ln 121 generally the word “mental” might be taken as pejorative perhaps “psychological” is a better term.
You have an interesting section on the use of cannabis and I suspect your University has significant experience here. You don’t discuss “doses” although I accept this is difficult. You could explore more the differences between CBD and THC and combinations thereof in treatment. Are there subsets of patients with FM who might benefit over others. Do you use THC in patients with “psychosomatic” FM.
Conclusions
Ln253
You return to your three channel concept. For treatment this does work (in my opinion) as you have described Psycholgical, physical and pharmacological treatments. However you return to the psychosomatic idea as well.
In your conclusion it would be helpful for readers to have your recommendations after your Literature review. Perhaps an algorithm to guide readers?
Reviewer 2 Report
New Insights about Fibromyalgia (FM) Syndrome Treatment
In general:
This is a well written review on the available treatment modalities of the FM symptoms.
The title is: “New” insights. I think the word “new” should be omitted, because all these therapy modalities and their effects are well known and the insights are not new.
Line 9 and line 51: “The symptoms of the disease also affect the patient mentally mainly due to the transparency” and “FM is a transparent disease, that is, a disease with no visible symptoms that mainly affects women.”
What do you mean by “transparency”?
The definition of transparency: Transparency in a process involves it being completely visible and open to scrutiny, so that it's clear that nothing is being hidden.
Fibromyalgia is rather the opposite of a transparent disease. The condition is completely invisible to family friends and colleagues. Also, the true pathophysiology of fibromyalgia is still “hidden” to physicians and scientists.
Line 11: “A typical symptom that characterizes FM patients is the lack of acceptance of the disease and hence the deficiencies in its management.”
I believe the deficiencies in its management are mainly due to the fact that the pathophysiology is not elucidated and therefore, targeted management is inexistant.
Hence, the results of therapies to improve the patients symptoms and their quality of life are often disappointing.
Line 42: “Since FM is associated with chronic pain without any visible damage to peripheral tissue,”
This claim is not correct. There is damage to the small fibers in up to 50% of patients with FM (numerous studies and reviews i.e. (Galosi, Truini and Di Stefano 2022) and even to the large fibers in a significant proportion of FM patients. (Caro, Galbraith and Winter 2018, Hulens et al. 2020) (Lawson et al. 2018) (Collado et al. 2021)
So, there is probably more to it than just “nociplastic” pain. Actually, the theory of central sensitization is just a hypothesis, which has not yet been indisputably substantiated. “The structural and functional alterations in the brain in small fiber neuropathy patients are more likely associated with sensory deafferentiation secondary to intra-epidermal depletion rather than sensitization.“ (H 2019)
FM is considered a neurological disorder
Line 83: Is 10 patients a sufficient number to draw any conclusions?
Line 106: “At the end of the program…”
Can you tell us how many patients were involved, inclusion and exclusion criteria, and how many dropouts were registered? Was the disease severity assessed pre-trial?
Line 123: “Various systematic reviews have [26,27] suggested that CBT is an effective approach to chronic pain management.“
Actually, only weak evidence for cognitive behavioural therapy and mindfulness was found in a review of studies and Eular recommendations of 2017 (Macfarlane et al. 2017)
Line 174: Physical Activity
It is obvious that engaging in (adapted) physical activity programs is beneficial for FM. However, in such programs, the worst FM cases are not reached. The proportion of patients that are too severely debilitated are unable engage in most exercise programs.
Also, the moderate or milder cases of FM tend to drop out less frequently.
Can you tell something about the severity of the FM cases involved in the studies and about the proportion of dropouts?
Line 193: “In addition, the effect of the disease also decreased between 5.3% 193 and 17.9%, while improving the symptoms of these patients”
What do you mean by the effect of the disease? Please specify.
Line 233: “THC is a psychoactive component that affects pain (as well as emotions) and works through… “
Indeed, I believe that Cannabinoids are suitable for the treatment of FM pain. However, the reader should be aware of the dangers of long term THC use: “Despite its therapeutic potential, a significant subpopulation of frequent cannabis or THC users will develop a drug use syndrome termed cannabis use disorder. Individuals suffering from cannabis use disorder exhibit many of the hallmarks of classical addictions including cravings, tolerance, and withdrawal symptoms.” (Kesner and Lovinger 2021)
Line 249: “As a somatic disease without evidence of threat or tissue damage or evidence of any causative disease”
Same remark. There is evidence of tissue damage i.e. small and large fibers.
Line 252: “In other words, the pain is the disease itself.”
You cannot state this as the true pathophysiology is not clarified.
Caro, X. J., R. G. Galbraith & E. F. Winter (2018) Evidence of peripheral large nerve involvement in fibromyalgia: a retrospective review of EMG and nerve conduction findings in 55 FM subjects. European Journal of Rheumatology, 5, 104-110.
Collado, A., J. Rivera, C. Alegre & B. Casanueva (2021) Fibromyalgia. Old opinions versus new knowledge. Reumatol Clin (Engl Ed), 17, 554.
Galosi, E., A. Truini & G. Di Stefano (2022) A Systematic Review and Meta-Analysis of the Prevalence of Small Fibre Impairment in Patients with Fibromyalgia. Diagnostics (Basel), 12.
H. 2019. Small Fiber Neuropathy and Related Syndromes: Pain and Neurodegeneration. Springer Singapore.
Hulens, M., F. Bruyninckx, R. Rasschaert, G. Vansant, P. De Mulder, I. Stalmans, C. Bervoets & W. Dankaerts (2020) Electrodiagnostic Abnormalities Associated with Fibromyalgia. J Pain Res, 13, 737-744.
Kesner, A. J. & D. M. Lovinger (2021) Cannabis use, abuse, and withdrawal: Cannabinergic mechanisms, clinical, and preclinical findings. J Neurochem, 157, 1674-1696.
Lawson, V. H., J. Grewal, K. V. Hackshaw, P. C. Mongiovi & A. M. Stino (2018) Fibromyalgia syndrome and small fiber, early or mild sensory polyneuropathy. Muscle Nerve, 58, 625-630.
Macfarlane, G. J., C. Kronisch, L. E. Dean, F. Atzeni, W. Häuser, E. Fluß, E. Choy, E. Kosek, K. Amris, J. Branco, F. Dincer, P. Leino-Arjas, K. Longley, G. M. McCarthy, S. Makri, S. Perrot, P. Sarzi-Puttini, A. Taylor & G. T. Jones (2017) EULAR revised recommendations for the management of fibromyalgia. Ann Rheum Dis, 76, 318-328.
